

# Effect of concurrent training on trainability performance factors in youth elite golf players

Juan Carlos Redondo[1], Ana María de Benito[2] and José María Izquierdo[1]

[1] Department of Physical Education and Sports, University of León, León, Spain
[2] Physical Activity and Sports Sciences Faculty, Catholic University of Valencia San Vicente Mártir, Valencia, Spain

## ABSTRACT

**Background**. Due to the early specialization of golf players, examining the within session sequence of training should be considered to enhance performance and prevent injury risk. The present study analyzed the effects of an 18-week concurrent training developed before or after a specific golf session in adolescence elite golfers on several performance factors.

**Methods**. Sixteen right-handed male golfers, were randomly divided into two groups: after golf specific training (AG) ($n = 8$, age: $16.77 \pm 0.58$ years) and before golf specific training (BG) ($n = 8$, age: $16.93 \pm 0.59$ years). AG and BG players followed a concurrent physical conditioning program (CT) after or before the golf specific training, respectively. Body mass, body fat, muscle mass, jumping ability (CMJ), ball speed (Sball), golf movements screens (GMS), power in a golf swing-specific cable woodchop (Wmax) and the perceived training load (TL) in golf specific training (TL-G) and TL in CT (TL-CT) were measured on three separate occasions.

**Results**. BG demonstrates a lower TL-CT than AG ($p < .001$, $\eta_p^2 = 0.90$) along the training program without effects on TL-G, achieving significant percentage of change on CMJ ($9.38\%$; $p = .165$; $d = 0.73$), GMS ($50.52\%$; $p = .41$, $d = 0.91$), Wmax ($16.93\%$; $p = .001$; $d = 2.02$) and Sball ($1.82\%$; $p = .018$; $d = 0.92$) without interaction effects on anthropometric measures.

**Conclusions**. Performing CT sessions before the regular golf training can improve specific performance factors with a lower perceived TL than the same training carried out after the regular golf training.

## INTRODUCTION

Golf is a skill-based sport (*Smith, Calliste & Lubans, 2011*) but also a demanding physical game with high power requirements (*Wells, Elmi & Thomas, 2009*). Thus, golf practitioners need an adequate physical condition, where combined strength training is deemed necessary to golf performance (*Doan et al., 2006*; *Thompson & Osness, 2004*).

In accordance with *Lloyd et al. (2015a)*, golf players begin to specialize at the end of adolescence period and very few of them become elite professional players. Therefore, the sport talent development is a core aspect both athletes and practitioners. Enhancing

Corresponding author
Juan Carlos Redondo,
jc.castan@unileon.es

youth golfers' performance is a complex and dynamic issue due to the varying interactions of growth, maturation, and training (*Lloyd et al., 2014*). For that reason, it is essential to design training strategies to optimize physical fitness and individual training response or trainability (*Hecksteden et al., 2015*). It is necessary to prescribe an accurate dosage of training load to prevent fatigue through training sessions and reduce injury risk. *Myer et al. (2011)* suggested that neuromuscular training is an effective method to prevent injuries in athletes when young.

In this sense, some studies conducted in youth golfers clarify the effects of strength training programs on physical fitness (*Alvarez et al., 2012*; *Lamberth et al., 2013*), but none has examined the within session sequence of neuromuscular training and sport-specific training as *Fernandez-Fernandez et al. (2018)* conducted in youth tennis players. These authors based their work on *Leveritt & Abernethy (1999)* who concluded that an acute bout of high-intensity endurance exercise may inhibit performance in a subsequent bout of resistance activity.

To the authors' knowledge, there appears to be a lack of studies investigating the effects of different concurrent training on golf performance. Therefore, the purpose of this study was to assess the effects of an 18-week concurrent training developed before or after a specific golf session in elite adolescent golfers on several performance factors. We hypothesized that a concurrent training conducted before the specific golf training session would demonstrate greater increases on performance factors than the same concurrent training conducted after the specific golf training session.

## MATERIALS & METHODS

### Study design

A parallel, 2-group, longitudinal study was designed to investigate the effects of two different approaches of training on selected golf performance factors. Selected subjects had similar handicap to avoid golf swing technical differences. We assigned volunteers to either a training group conducting a concurrent physical conditioning program (CT) before golf specific training (BG) or a group that performed CT after golf specific training (AG). After a familiarization period, laboratory tests, and a specific range of physical-performance, participants were evaluated on three occasions; 1 week before the start of the training program (T1), after 12 weeks of training (T2) and after 18 weeks of training (T3). Also, subjects reported to be free from any injuries, surgeries or sport related rehabilitation during the 12 months prior to starting the study. The flowchart for recruitment and testing is displayed in Fig. 1.

The research was conducted during the competitive season (i.e., February, March, April, May and June). Two months before the beginning of the study participants conducted the same regular golf training program. Participants were instructed for not alter their lifestyle during the investigation period in order to reduce potential interference. They were not allowed to exercise or consume stimulant drinks at least 24 h prior to test.
## Study FlowChart

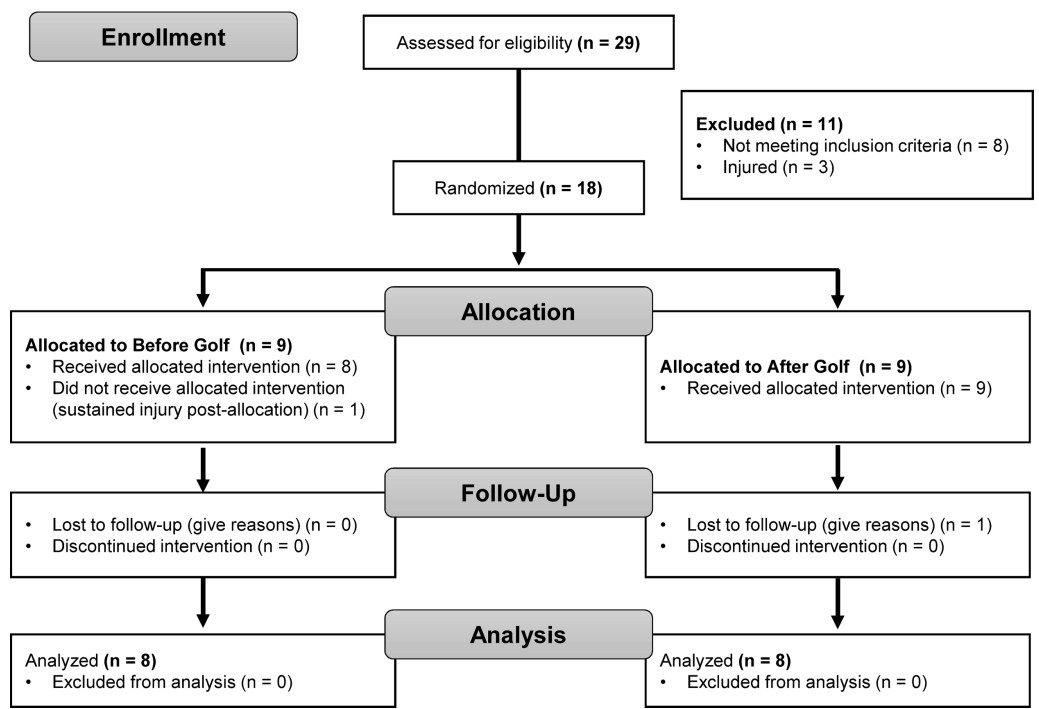

**Figure 1** Recruitment and testing flowchart of participants through the intervention.

**Table 1** Demographic and anthropometric data of the players (mean ± SD).

| Group | $n$ | Age (y) | Experience (y) | Handicap | Height (cm) | Mass (kg) | PHV |
|-------|-----|---------|----------------|----------|-------------|-----------|-----|
| AG | 8 | 16.44 ± 0.67 | 3.1 ± 1.2 | 0.24 ± 0.79 | 176.14 ± 6.98 | 71.56 ± 7.81 | 1.84 ± 0.70 |
| BG | 8 | 16.28 ± 0.58 | 3.0 ± 0.9 | 0.34 ± 1.21 | 176.28 ± 4.08 | 67.38 ± 12.41 | 1.53 ± 0.80 |

**Notes.**

AG, after golf specific training; BG, before golf specific training; PHV, peak height velocity.

## Participants

Based on the previous study by *Alvarez et al. (2012)* a priori power analysis (G*Power3) with $\alpha < 0.05$ and $1-\beta = 80$ indicated that a sample size of at least 14 was required to explore the differences between sequencing effects of neuromuscular training. A total of 16 elite right-handed youth male golfers voluntarily agreed to participate in the study and were randomly divided into two groups: before golf specific training (BG, $n = 8$) and after golf specific training (AG, $n = 8$). There were no group differences ($p > .05$) with regard to demographic and anthropometric data showed in Table 1. Players averaged $9.4 \pm 0.9$ h of training per week and completed at least one full round of golf per week.

All players involved in the study attended all the sessions. Legal guardians and all participants were provided an explanation of testing and training protocols and they gave written informed consent prior to data collection. They also completed a set of

questionnaires on their health history and golf-playing history. This study was conducted in accordance with the guidelines found in the Declaration of Helsinki and all procedures involving human subjects were approved by the University of León Ethics Committee (ULE2018-2019-76).

## Testing procedures

### Anthropometric data

Anthropometric testing followed the International Society for the Advancement of Kinanthropometry protocols (ISAK). Fat mass, residual mass, bone mass, and muscle mass and their respective percentages were computed to estimate body composition; *Deurenberg, Weststrate & Seidell, 1991*.

In order to estimate the maturity status of participants, the peak-height-velocity (PHV) was calculated according to *Mirwald et al. (2002)*. All anthropometric measures were highly reliable with intraclass correlation coefficients (ICCs) of 0.91 to 0.98 for skinfolds and 0.93 to 0.98 for diameters.

### Training load quantification

During 18 in-season weeks, the perceived training load (TL) was quantified using the session rating of perceived exertion (sRPE) method (*Foster et al., 2001*). Ten minutes after each training and using Foster's 0–10 scale (*Foster et al., 2001*), participants were asked by the same person (fitness coach) on all occasions to rate their general perception of the session difficulty (PE) (*Chen, Fan & Moe, 2002*). We allowed players to mark a plus sign (interpreted as 0.5 points) alongside the integer value (*Otaegi & Los Arcos, 2020*). All the golfers were familiarized with this method during the previous months. All golf specific training and CT PEs recorded during the study were summed separately. Then, TL was calculated by multiplying the PE value by the duration of the training. Partial 12-weeks (T2) and 18-weeks (T3) TL in golf specific training (TL-G) and TL in CT (TL-CT) were considered for each group (*Otaegi & Los Arcos, 2020*). The duration of a training session (training volume) was recorded for each player from the start to the end of the session, including recovery periods but excluding stretching exercises (*Los Arcos et al., 2015*).

### Golf movement screen

We applied a specific golf movement screen (GMS) to examine the movement competency of golf players. According to *Gulgin, Schulte & Crawley (2014)*, the subjects performed 13 different tests (movement screens). These tests, established by the Titleist Performance Institute (TPI), provide data with respect to stability, mobility, coordination of body segments, and balance. The sum of the 13 GMS was recorded. The ICC was 0.98.

### Lower limbs explosive strength

Golfers performed a countermovement jump (CMJ) without arm swing on a jumping mat (SportJUMP System; DSD, Spain) according to *Bosco, Luhtanen & Komi (1983)*. Golfers performed two maximal CMJs intercalated with 60 s of passive recovery. Only the best height for each participant was recorded. The ICC of the CMJ was 0.97 and the CV was 4.1%.

### Rotational golf-specific exercise

According to *Keogh et al. (2009)* the golf swing-specific cable woodchop (GSCWC) is a rotational exercise that is very similar to the golf swing in terms of posture, range of motion, intended velocity, direction of force (torque) application, and coordination patterns.

A one-repetition-maximum (1-RM) test following the protocol established by the National Strength and Conditioning Association was performed to measure peak power output (highest instantaneous value) during each GSCWC exercise. The peak power outputs (Wmax) expressed in watts were measured with a pneumatic resistance device (Infinity, Keiser, Calif. USA) according to *Peltonen, Häkkinen & Avela (2013)*. The ICC was 0.99.

### Driving performance

Ball speed (Sball) was assessed using new regulation golf balls (Titlest Pro V1, USA), and tees of various heights to suit the preference of each participant. According to *Alvarez et al. (2012)*, Sball expressed in km h-1 was measured with a Stalker's type hyperfrequency radar (Stalker Professional Radar, Radar Sales, Plymouth, MA, USA). Each participant performed five drives at the maximum speed possible using his own club. The ICC for this test was 0.94.

### Training intervention

During the 18-week intervention, golfers carried out four training sessions per week: two CT sessions (on Wednesday and Friday), one putter-and-approach session on Monday and one full round of golf (on Saturday or Sunday).

The CT program based on a mix of golf-specific functional movement training and neuromuscular training program (Table 2) was undertaken at an indoor facility (high performance sports center) on average $66.70 \pm 3.1$ min per session. The regular golf training took place at an outdoor facility (sport golf club center) located 30 min from each other. According to *Fernandez-Fernandez et al. (2018)* during recovery period between training bouts, all participants could ingest water and carbohydrate/electrolyte drink. Regular golf training lasted on average $83.2 \pm 9.6$ min and was characterized by a $\sim$10-minute specific warm-up (i.e., general mobility and low-intensity golf shots), $\sim$30 min of technical swing adjustments, and $\sim$40 min of specific drills (i.e., mixed iron-drives-putter drills).

All players had previous experience of this type of training. Prior to starting CT, participants performed a standardized 10-minute warm-up protocol. After the warm-up, each golfer developed a 30-minutes personal golf-specific functional movement training program with conditioning exercises designed to enhance the lower body stability and the upper body mobility (*Lephart et al., 2007*). Lastly, participants proceeded with the neuromuscular training program divided into three parts (maximal strength; explosive strength and golf-specific strength training), each six weeks long (*Alvarez et al., 2012*). Details are given in Table 2.

## Data analyses

The data were checked for normality using the Shapiro–Wilk test and found to be suitable for parametric testing. Student's t-tests were performed to determine differences between

**Table 2  Neuromuscular training details.**

| Maximal Strength Training | |
| --- | --- |
| Resistance Exercise | Sets/Repetitions/Load/Rest Period Between Sets |
| Horizontal bench press | 3 sets × 5 repetitions × 80%/4 min |
| Seated row machine | 3 sets × 5 repetitions × 80%/4 min |
| Leg press machine | 3 sets × 5 repetitions × 80%/4 min |
| Seated calf extension | 3 sets × 5 repetitions × 80%/4 min |
| Triceps cable push-down | 3 sets × 5 repetitions × 80%/4 min |
| **Explosive Strength Training** | |
| Combined exercise | Sets/Repetitions/Load/Repetitions/Rest Between Sets |
| Horizontal bench press + plyometric push-ups | 3 sets (6 repetitions × 70% + 10 repetitions)/4 min |
| Seated row machine + explosive pull-downs | 3 sets (6 repetitions × 70% + 10 repetitions)/4 min |
| Leg press machine + vertical jumps over hurdles (45 cm) | 3 sets (6 repetitions × 70% + 10 repetitions)/4 min |
| Seated calf extension + vertical jumps over hurdles (45 cm) | 3 sets (6 repetitions × 70% + 10 repetitions)/4 min |
| Triceps cable push-down + plyometric push-ups | 3 sets (6 repetitions × 70% + 10 repetitions)/4 min |
| **Golf-Specific Strength Training** | |
| Exercises | Sets/Repetitions/Rest Between Sets |
| Golf drives with weighted clubs | 3 sets × 10 repetitions/4 min |
| Accelerated drives with an acceleration tubing club system | 3 sets × 10 repetitions/4 min |

groups at baseline. A 2x3 repeated measures ANOVA was used to explore the effects of group (AG and BG), and time (one week before training, 12 weeks and 18 weeks after training). When a significant $F$ value was achieved by means of Wilks' lambda, Scheffe's post hoc procedures were performed to locate the pairwise differences. In addition, partial eta squared ($\eta_p^2$) was computed to determine the effect size which was interpreted as small 0.1, medium 0.3, and large 0.5. The percentage difference between groups was assessed using one-way ANOVA by comparing T1–T2, T1–T3 and T2–T3 and the Cohen's $d$ (*Cohen, 1988*) was calculated to determine the magnitude of differences between experimental conditions for each variable. The significance level was set at p ≤ .05. Statistical analysis was performed with SPSS 24.0 (IBM® SPSS Statistics 24, IBM GmbH).

## RESULTS

Overall, golfers completed more than 95% of the training sessions, proving a very good adhesion to the training program. Student's $t$-test between AG and BG at baseline reveled that there were no statistically significant differences ($p > .05$) before the start of the training program with regard to the analyzed variables. The mean and standard deviation and main effects for the different variables are reported in Tables 3 and 4.

### Anthropometric data
ANOVA revealed no significant time x group interaction for anthropometric measures, although significant improvements were seen between the time points for both groups. Further post hoc analysis showed significant increase of body mass between T1 and T2 ($p < .001$; $d = 0.14$), T2 and T3 ($p = .03$; $d = 0.04$) and T1 and T3 ($p = .002$; $d = 0.18$) in

**Table 3 Descriptive and inferential anthropometric results from 2 (group) × 3 (time) ANOVA.**

| Group/time | | Body mass, kg | | Body fat percent | | Muscle mass percent | |
|---|---|---|---|---|---|---|---|
| | | M | SD | M | SD | M | SD |
| AG | T1 | 71.56 | 7.81 | 10.45 | 2.13 | 48.28 | 2.01 |
| | T2 | 72.25[**] | 7.31 | 10.37 | 2.01 | 48.53 | 1.97 |
| | T3 | 72.71 | 7.71 | 10.11 | 2.13 | 48.68 | 1.98 |
| BG | T1 | 67.38[*] | 12.41 | 11.42 | 2.58 | 46.87 | 1.94 |
| | T2 | 69.01[**] | 11.86 | 11.55 | 2.24 | 47.18 | 1.84 |
| | T3 | 69.51 | 11.77 | 11.01 | 2.11 | 47.47 | 1.63 |
| **RM ANOVA** | | $p$ | $\eta_p^2$ | $p$ | $\eta_p^2$ | $p$ | $\eta_p^2$ |
| Group | | .49 | 0.03 | .36 | 0.06 | .18 | 0.12 |
| Time | | <.001 | **0.57** | .20 | 0.2 | .001 | 0.42 |
| Time × Group | | .15 | 0.13 | .84 | 0.01 | .65 | 0.03 |

**Notes.**

AG, after golf group; BG, before golf group; T1, 1 week before training program; T2, after 12 weeks of training; T3, after 18 weeks of training; $p$, $p$ value; $\eta_p^2$, effect size.

[*]Significant difference between T1 and T2.

[**]Significant difference between T2 and T3.

**Table 4 Descriptive and inferential perceived training load and performance results from 2 (group) × 3 (time) ANOVA.**

| Group/time | | TL-CT, au | | TL-G, au | | CMJ, cm | | GMS, au | | Sball, km h⁻¹ | | Wmax, w | |
|---|---|---|---|---|---|---|---|---|---|---|---|---|---|
| | | M | SD | M | SD | M | SD | M | SD | M | SD | M | SD |
| AG | T1 | n/a | | n/a | | 40.15[*] | 4.98 | 11.13 | 4.42 | 256.63 | 8.63 | 965.13[*] | 204.57 |
| | T2 | 349.06 | 4.57 | 209.27 | 5.78 | 42.13[**] | 4.91 | 8.75 | 5.73 | 258.5 | 6.65 | 1010.88[**] | 209.19 |
| | T3 | 90.63 | 6.65 | 210.0 | 10.34 | 42.29§ | 4.88 | 7.63 | 4.07 | 259.5§ | 6.85 | 1048.88§ | 208.49 |
| BG | T1 | n/a | | n/a | | 37.07[*] | 5.11 | 13.5[*] | 5.73 | 250.88[*] | 15.3 | 992.75[*] | 164.53 |
| | T2 | 325.07 | 6.51 | 211.46 | 5.17 | 38.79[**] | 4.89 | 8.5[**] | 3.46 | 254.5 | 14.97 | 1102.0[**] | 160.93 |
| | T3 | 81.25 | 7.32 | 213.65 | 10.31 | 40.88 | 4.64 | 5.88 | 1.64 | 256.63 | 14.99 | 1193.50 | 168.32 |
| **RM ANOVA** | | $p$ | $\eta_p^2$ | $p$ | $\eta_p^2$ | $p$ | $\eta_p^2$ | $p$ | $\eta_p^2$ | $p$ | $\eta_p^2$ | $p$ | $\eta_p^2$ |
| Group | | <.001 | **0.79** | .46 | 0.04 | .30 | 0.08 | .16 | 0.25 | .45 | 0.04 | .36 | 0.06 |
| Time | | <.001 | **0.99** | .26 | 0.19 | <.001 | **0.65** | <.001 | **0.64** | <.001 | **0.63** | <.001 | **0.87** |
| Time × Group | | .005 | 0.45 | .86 | 0.02 | .06 | 0.18 | .02 | 0.25 | .15 | 0.13 | <.001 | **0.53** |

**Notes.**

n/a, not applicable; AG, after golf group; BG, before golf group; T1, 1 week before training program; T2, after 12 weeks of training; T3, after 18 weeks of training; $p$, $p$ value; $\eta_p^2$, effect size; TL-CT, perceived concurrent physical conditioning training load; TL-G, perceived specific golf training load; au, arbitrary units; CMJ, countermovement jump; GMS, golf movement screen; Sball, ball speed; Wmax, maximal power.

[*]Significant difference between T1 and T2.

[**]Significant difference between T2 and T3.

BG, and between T2 and T3 ($p = .04$; $d = 0.06$) in AG. Related to percent muscle mass, a significant increase was observed in BG between T1 and T3 ($p = .04$; $d = 0.34$).

## Perceived training load

Data analysis revealed significant time x group interaction effects just for TL-CT ($p = .005$; $\eta_p^2 = 0.45$). Differences between T2 and T3 were dismissed as non-logic (neuromuscular training load in T2 not comparable with golf-specific training load in T3).

## Performance variables

Analysis of variance located significant time x group interaction effects for GMS ($p = .02$; $\eta_p^2 = 0.25$) and Wmax ($p < .001$; $\eta_p^2 = 0.53$). Additionally, ANOVA revealed a significant effect for time in all the performance variables. Regarding to BG, Scheffe's post hoc tests located the differences between T1 and T2 differences were located in GMS ($p < .002$; $d = 1.06$), CMJ ($p < .001$; $d = 0.34$), Wmax ($p < .001$; $d = 0.67$) and Sball ($p < .01$; $d = 0.24$), between T1 and T3 in GMS ($p < .001$; $d = 1.81$), CMJ ($p < .001$; $d = 0.78$), Wmax ($p < .001$; $d = 1.21$) and Sball ($p < .001$; $d = 0.31$), and between T2 and T3 differences were located in GMS ($p = .003$; $d = 0.97$), CMJ ($p = .03$; $d = 0.44$) and Wmax ($p < .001$; $d = 0.56$). Furthermore, post hoc analysis for AG located the differences between T1 and T2 differences were located in CMJ ($p < .001$; $d = 0.40$) and Wmax ($p < .001$; $d = 0.22$), between T1 and T3 in CMJ ($p = .02$; $d = 0.43$), Wmax ($p = .002$; $d = 0.41$) and Sball ($p = .02$; $d = 0.34$), and between T2 and T3 differences were located in CMJ ($p = .02$; $d = 0.03$) and Wmax ($p = .03$; $d = 0.18$).

With regard to comparison of the percentage of change between evaluations (T1, T2 and T3) in association with the TL data are represented in Fig. 2. Concerning TL-CT, one-way ANOVA revealed a significant effect between AG and BG in T2 ($p < .001$; $d = 3.43$) and T3 ($p = .018$; $d = 1.34$). T1–T3 comparison between groups shows that BG obtains higher percentages of change in all performance variables: CMJ (AG +37.15%; BG +50.52%; $p = .041$; $d = 0.91$), GMS (AG +5.08%; BG +9.38%; $p = .165$; $d = 0.73$), Wmax (AG +8.03%; BG +16.96%; $p = .001$; $d = 2.02$) and Sball (AG +1.03%; BG +1.82%; $p = .018$; $d = 0.92$).

# DISCUSSION

The purpose of the current investigation included comparing the effects of an 18-week concurrent training developed before or after a specific golf session in adolescent elite golfers on several performance factors. Key findings for the sequencing effects of training programs showed that CT conducted before the specific golf training demonstrate greater increases on performance factors and less perceived training load than the same CT conducted after the specific golf training.

Experts have considered the use of session RPE needless for on and off-course golf activities and it would seem that rejection is based on a perceived low intensity of golf (*Williams et al., 2018*). However, such considerations have not been raised into the context of routinely training programs of golf players. In regular golf training, BG and AG were requested to execute a wide range of skills that require both fine motor skills and muscular power over an extended period of time (*Hellström, 2009*). In this context, TL-G outcomes showed no differences for BG compared to AG. Nevertheless, the results in TL-CT for AG ($321.57 \pm 3.34$ in T1 and $376 \pm 5.87$ in T2) showed values significant different from BG ($325.07 \pm 6.51$ in T2 and $81.25 \pm 7.32$ in T3). This would indicate that golfers in the BG were carrying out the CT sessions in a less sense of fatigue, resulting in greater increases on performance factors possibly due to a less stress (*Blume et al., 2018*).

Improvements obtained in CMJ (5.08% for AG and 9.38% for BG) are agree with previous studies (*Alvarez et al., 2012*; *Driggers & Sato, 2018*; *Kenny et al., 2017*), confirming

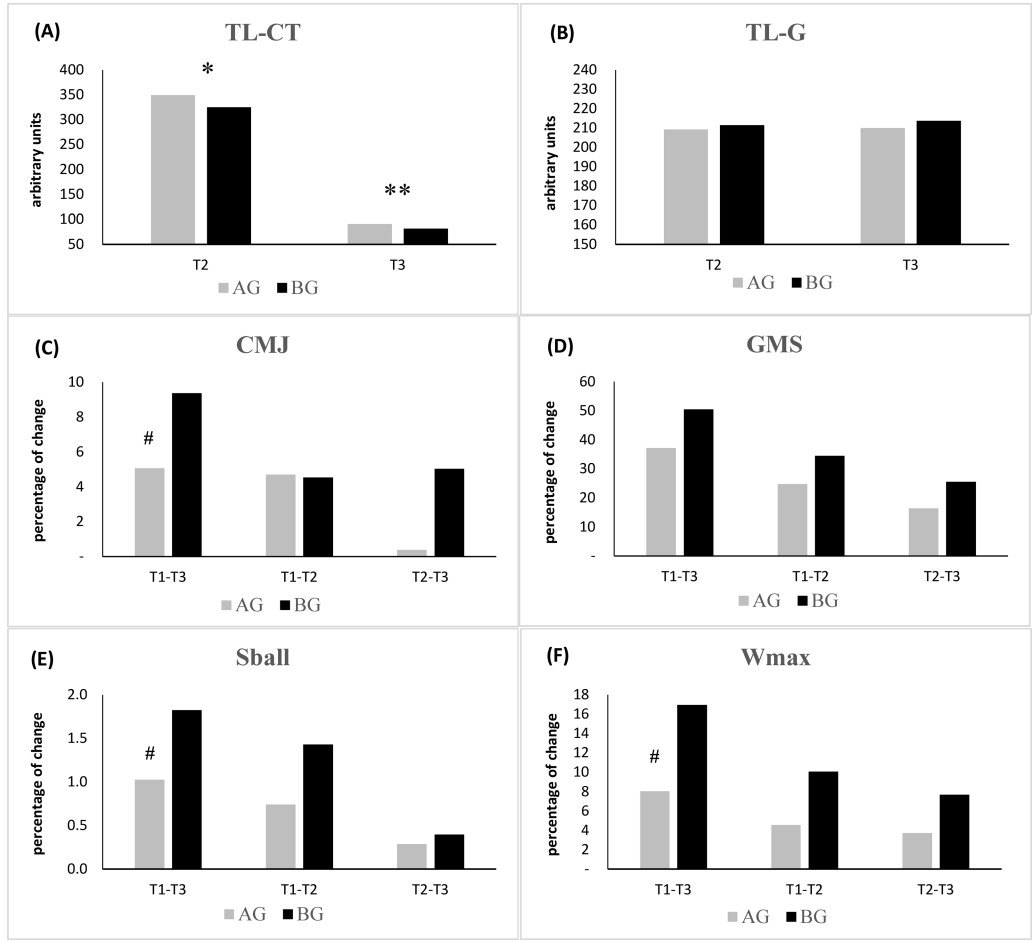

**Figure 2** **Comparison of the percentage of change between evaluations (T1, T2 and T3).** AG, after golf group; BG, before golf group; au, arbitrary units; (A) TL-CT, perceived concurrent physical conditioning training load; (B) TL-G, perceived specific golf training load; (C) CMJ, countermovement jump; (D) GMS, golf movement screen; (E) Sball, ball speed; (F) Wmax, maximal power. #percentage of change larger in BG than in AG ($p < .001$); *significance difference between AG and BG ($p < .001$).

that a twice-weekly strength training program, using the protocol outlined, was associated with enhancements in driving performance (*Wells, Elmi & Thomas, 2009*), lower limb explosive strength and rotational power. Our results revealed that golfers transfer the gains to the driver performance with percentage improvement ranging from 1.03% to 1.82% for AG and BG respectively.

Previous studies have reported the positive influence of strength training on driving performance in highly-trained players (*Alvarez et al., 2012*; *Driggers & Sato, 2018*; *Fletcher & Hartwell, 2004*). This is confirmed by the results of our study, which proves that the CT program followed increase driving performance, both Sball and rotational power. Our results show that BG obtains higher improvements than AG in GMS (50.52% vs 37.15%) and rotational power (16.96% vs 8.03%) while maintaining similar differences in Sball. The enhancements in driving performance could be related to improvements in GMS such

as stability, mobility, body segment coordination and balance (*Gulgin, Schulte & Crawley, 2014*; *Myers et al., 2008*; *Speariett & Armstrong, 2019*). Further 3D motion analysis work would provide deeper analysis to clarify the relation between swing mechanics and strength and golf movement screens.

Golf can be one of those sports traditionally favor early specialization (*Lloyd et al., 2015b*) for this reason golfers should be engaged with an integrative strength and conditioning programs focused on diversifying motor skill development and enhancing muscle strength to maximize performance and reduce injury risk (*Faigenbaum et al., 2014*). In this regard, our data show that concurrent training programs combining physical conditioning program and golf specific training may have an important impact on performance factors such as drive ball speed (*Torres-Ronda, Delextrat & Gonzalez-Badillo, 2014*) or CMJ (*Driggers & Sato, 2018*; *Kenny et al., 2017*). In addition, session sequence should be considered since our results suggest that performing strength training before golf-specific work allows golfers get a better or similar performance with a lower perceived TL (9% lower BG than AG) which supports the results of 18% obtained by *Fernandez-Fernandez et al. (2018)*.

## CONCLUSIONS

On the basis of our results, it may be concluded that implement concurrent training before a specific-golf session in young golf players is more effective over golf performance factors (e.g., jumping performance, ball speed, rotation power). Thus, coaches would develop combined golf-specific functional movement (e.g., Titleist Performance Institute, level 1 golf fitness screen) and neuromuscular training program divided into three parts (maximal, explosive and golf-specific strength). From a practical point of view, CT sessions should not exceed a total volume of 70 min (including the warm- up) and an appropriate resting time before the following golf training should be above 30 min.

### Funding
The authors received no funding for this work.

### Competing Interests
The authors declare there are no competing interests.

### Author Contributions
- Juan Carlos Redondo conceived and designed the experiments, performed the experiments, analyzed the data, prepared figures and/or tables, authored or reviewed drafts of the paper, and approved the final draft.
- Ana María de Benito performed the experiments, prepared figures and/or tables, authored or reviewed drafts of the paper, and approved the final draft.
- José María Izquierdo performed the experiments, analyzed the data, prepared figures and/or tables, authored or reviewed drafts of the paper, and approved the final draft.

## Ethics

The following information was supplied relating to ethical approvals (i.e., approving body and any reference numbers):

This study was conducted according to the guidelines laid down in the Declaration of Helsinki and all procedures involving human subjects were approved by the University of León Ethics Committee (ULE2018-2019-76).

## Data Availability

The raw measurements and the codebook for the data are available as Supplemental Files.

## Supplemental Information

Supplemental information for this article can be found online at http://dx.doi.org/10.7717/peerj.9963#supplemental-information.

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
