# Peer review of "Effect of concurrent training on trainability performance factors in youth elite golf players"

_PeerJ, doi:10.7717/peerj.9963_

## Round 0.1 · original submission · Major Revisions

Reviewers saw merits in the study. However, some changes should be implemented to improve the manuscript. I look forward to receiving your submission.

Reviewer 1 ·

Basic reporting

The writing style is clear to begin with but seems to lose clarity as the paper goes on. The figures need to be cleared up. The introduction provides an adequate attempt to rationalise the study but there are a numerous elements which need to be clearer.

Experimental design

Research question is clearly defined. And there appears to be a gap in the literature. The methods section needs a lot of work to attain the clarity that is needed.

Validity of the findings

It is hard to comment on the validity of the findings because the methods and results are not very clear. See below for detail.

Additional comments

Abstract
The section requires a background sentence, not just the aim.
Also, in the results there is no mention of the other outcome variables.

Introduction
The rationale is built up fairly. It would be worth trying to improve the coherency of this however. At times the sentences ‘jump’ about.
Also, you need to rationalise the key performance indicators of golf. You measure CMJ, ball speeds etc but there is no information for the reader on why these are important. I believe Jack Wells and Danny Coughlan have done some work on this with golfer from the challenge tour. You might look to his work.
Need to include a rationale for why sequencing is important. This could be from an interference effect perspective or from a fatigue management perspective.

Methods
The study section is really haphazard. You need to structure this better and condense it so it is easier to read.
Table 1 – you need to t test these to see if there are any differences between groups for these characteristics. It looks like there might be for mass and PHV
PTL isn’t a standard abbreviation as I’m aware. Also, and this is really pedantic, but the underscore makes this hard to really. Normally a hyphen is used e.g. TL-golf or TL-G. It looks like a code for an SPSS variable otherwise
If you’re measuring ball speed, did the participants use the same drivers?
L136 – should be ‘a specific golf movement’
L143 – the ICC from the is study or a previous report?
GSCWC – you need to state the outcome metrics here. Also, you include a rationale for this test. This is a method section so you don’t need to do this here in my opinion
1RM of what exercise? You also not got a subsection for this. Its not very clear that you’re measuring two things here, max strength and optimal power
L188 make reference to the table with the programme in it
Why Kolmogorov-Smirnov instead of Shapiro-Wilk? You sample is only 16


Result
You’ve not analysed your flow chart so put this in the method
L218 a similar what? This is a bit lazy. You need to describe your data
L222 you need to report the statistics from the ANOVA i.e. the main and interaction effects
L225 please include effect size
L229 what do you mean non-logic?
There is too much to comment on in this results section. You need to report the statistics clearly; include the ANOVA results, than the post hoc test where necessary and the effect sizes. Its not clear if the effect sizes are between or within group or the time points. Given the volume of data you have you need to write this in that highlights the differences between groups.
Why have you reported percentage changes in the graphs and not the raw values?

Discussion
L251 You’ve not reported any perceived fatigue data
L253 this paragraph makes no attempt to explain your data. Remove.
L258 I think you’ve measured power and not strength? This isn’t entirely clear
L266 ‘there are many researches’ doesn’t make sense
L273 instead of providing a direction for future work here, comment on the practical implications of your work
L280 this is pedantic, but you’ve put data for percentages, so you can’t say mass. You might report the raw fat and muscle mass data. This is especially important because body mass has changed.
Need to elaborate on the Fernandez-Fernandez study because this detail isn’t clear.
L296 the sRPE scale which you used measures exertion, not fatigue. There is a massive difference. Yes the two are associated in some instances, but you’ve had not measured them in this manner.
L297 this is not a validity study. This is also a very strange paragraph.
In this section you need to discuss and explain your data. There is very little attempt to do this. It needs a wholesale change.

·

Basic reporting

Some English fixes are necessary
Line 50: "it is necessary to...."
Line 59 & 64: the following words are jargon, erase "In this case" "consequently"

In introduction, authors should have specific rationale relating to study purpose and hypothesis. For example, why comparing before and after golf specific training? In the introduction, there is no information about it. Line 56-59 mentioned very little specific about it relating to fatigue. Maybe because it is junior level?

Line 151-165: I believe figures of start and end position seems much easier to understand rather than descriptive. Do you have a picture of the exercise movement?

Line 258: "improvements in muscular strength were shown between"

In discussion, it is documented that concurrent training before golf specific training works better. However, authors should also mention that this is junior elite level golfers. when they are matured, maybe the case would be different, means it would NOT matter when to do training because high threshold of physical activity, OR because golf performance is optimal, very little improvement in performance measure could be captured. Also, what could happen to non-elite level junior golfers. Obviously, there are far more non-elite than elite junior players. Would this data could be a good representation of improvement of performance measures from physical training? Those should be mentioned as leading to future studies.

Experimental design

Line96: when "randomly divided". specifically how? How did authors do randomization?

Validity of the findings

data seems accurate and easy to follow in the tables and figure. Very well constructed for data display

Additional comments

The purpose of the study was to investigate physical training helps junior golfers to physical gain and performance measures for golf. The paper is very well constructed and written. Some minor edits are necessary but overall very nice approach to identify the results.

---

## Round 0.2 · Major Revisions

Some issues are still present in the manuscript. The manuscript is genrally poorly formatted and this needs to be considered much more carefully to facilitate the review process.

Reviewer 1 ·

Basic reporting

See below comments

Experimental design

See below comments

Validity of the findings

See below comments

Additional comments

Introduction
- L40, it isn’t clear who ‘their’ is relating to. Please rephrase. Again, the coherency of this work needs to be improved. Even the next sentence, “In this sense, according to the scientific literature…” is quite unclear.
- L44 by definition of elite, few people become this. Can you allude to why this is different for golf?
- L47 split this sentence into two, it is very long
- L50 this sentence only focuses on injury, yet you write about fitness more generally beforehand
- As per my previous comments you still haven’t rationalise your key performance markers. Instead, the information is included within the discussion (L269). If all of these factors are explained in the Alvarez and Lamberth studies, then you need to provide more detail on them in the introduction. This is not satisfactory from the authors and my original comment still stand.

Methods
- The document that has been sent back is really poorly formatted and its quite hard to follow the methods
- The 1RMs for exercises are still unclear
- Amendments have been made adequately.

Results
- You still haven’t reported a lot of the main effects, only the interactions e.g. you haven’t reported a main effect for the anthropometric data of the perceived training load.
- Its unclear what neuromuscular training load is. The point of taking RPE is that it provides a global indication of the load. Also, what makes you think this is neuromuscular load?

Discussion
- The discussion is much improved, but needs to be read over. Some of the sentence, here and throughout, the manuscript require greater clarity

·

Basic reporting

The paper is much clear after revising the document. Because of major reviews, authors have minor editorial mistakes such as putting two periods in the statement which need to be addressed. Background information as well as relevancy of the study is well documented. Figures and Tables are clear but need to be corrected on table 1 (AG). the results seems to help practitioners to make necessary approach to train junior golfers.

Experimental design

The study purpose is very attractive to many practitioners and it is well designed with complex analysis. Approach to the problem has detailed information of data.

Validity of the findings

Validity of finding seems very good. However, I am concerned about how authors are now using eta squared as a definition of effect size. To my knowledge, SPSS software automatically provide eta squared for effect size, which is somewhat concerning giving a fact that eta squared effect size is only adequate in relationship study (using correlation). For comparison study such as this one, very common effect size method uses Cohen's D.

Additional comments

Here are some minor edits
Line 65 erase
Line 83 erase period
Table 1 handicap for AG error
Line 94 provide an explanation
Line 128 already abbreviated at Line 124
Line 133 spacing between "participant" and "was"
Line 142 erase period
Line 145 erase period

---

## Round 0.3 · accepted · Accept

Congratulations for meeting the high standard publications of PeerJ!

·

Basic reporting

no comment

Experimental design

no comment

Validity of the findings

no comment

Additional comments

The paper is a well written document